# How Did Sheep Save the Day? The Role of Dry Stone Wall Heritage and Agropastorality in Historical Landscape Preservation. A Case-Study of the Town of Cres Olive Grove

Tanja Kremenić [1,2], Goran Andlar [2,*] and Mauro Varotto [1]

[1] Department of Historical and Geographic Sciences and the Ancient World, University of Padova, 35141 Padova, Italy; tanja.kremenic@phd.unipd.it (T.K.); mauro.varotto@unipd.it (M.V.)

[2] Department of Ornamental Plants, Landscape Architecture and Garden Art, Faculty of Agriculture, University of Zagreb, 10000 Zagreb, Croatia

[*] Correspondence: gandlar@agr.hr; Tel.: +385-95-912-3055

**Abstract:** The dry stone wall landscape surrounding the town of Cres is a unique cultural landscape; it is one of the largest well-preserved historical olive groves in the Croatian Adriatic, while simultaneously serving as pasture for sheep. Still, there are currently no studies that capture this landscape as a multifunctional whole or which acknowledge its relevance within the current multidisciplinary discussions. The aim of this paper is to clarify past and current circumstances surrounding and affecting it. The study focuses on two main pillars of landscape preservation: (1) dry stone wall structures and (2) agro-pastoral practices; giving an overview of its historical formation and current management and trends. The goal is to establish knowledge that can be used as a foundation for the management of this area and present a good practice example for the preservation of historical landscapes in the Mediterranean region. The research involved combined desk and fieldwork: cartographic data analysis, literature analysis, GIS elaboration, terrestrial and aerial photographs and observations, followed by interviews with local informants. Continuous investing in the production of quality olive oil and the evolution of the landscape into a multifunctional agro-pastoral-touristic space is what enabled its preservation. This multifunctionality can only be matched by a diversity of scientific studies and this study aimed at providing the first step—a foundation for the identification of the values of the Cres landscape, with the scope of better precising further planning and management.

**Keywords:** dry stone walls; olive-sheep association; agro-pastoral system; cultural landscape; island of Cres; Croatia; Mediterranean



## 1. Introduction

Historical dry stone wall (agri)-cultural landscapes have been a subject of scientific appreciation for decades. They have been observed through the category of "terraced landscapes" on a global scale, forming a scientific-expert-enthusiast International Terraced Landscapes Alliance (ITLA) in 2010 [1,2], Mediterranean scale [3–6] and Croatian scale, focusing both on terraces [7] and dry stone wall enclosures and structures [8–14]. Institutional recognition of such landscapes, its tangible and intangible constituents, has been promoted by various organisations, programmes and initiatives: UNESCO [15,16], ELC [17], IUCN [18], GIAHS [19], PDO and PGI [20], and GAEC [21]:UNESCO was the first to protect "cultural landscapes" in 1972/1992 within the Convention concerning the Protection of the World Cultural and Natural Heritage; it supported agro-pastoral cultural landscapes and practices since 2010 and included the "art of dry stone walling, knowledge and techniques" in the Representative List of Intangible Cultural Heritage of Humanity (2018); ELC—European Landscape Convention of the Council of Europe (2000); IUCN—The International Union for Conservation of Nature giving particular emphasis on Protected landscape management category V as an integral framework for protection of

(rural and agri-) cultural landscapes; GIAHS-FAO's programme for "Globally Important Agricultural Heritage Systems" for the protection of outstanding landscapes representing aesthetic beauty, agricultural biodiversity, resilient ecosystems and cultural heritage (2002/2014); PDO—Protected Designation of Origin and PGI (Protected Geographical Indication) are quality marks for agricultural landscapes products (2012); GAEC-Good agricultural and environmental conditions, a set of EU Rural policy standards aiming to achieve a sustainable agriculture (2013), etc.

These cultural landscapes remain relevant due to their visual impressiveness and the fascination with human effort that the dry stone structures are a testament to, but also due to the difficulties in finding appropriate management models, which would avoid the post-productive path of degradation [22–26] and the disregard of their evolving and dynamic nature through museumification and commodification [27–31]. The resistance of this landscape's main constructive element—dry stone walls—results in a longer-lasting image of the agricultural landscape than in lowland continental landscapes. Ancient dry stone walled agricultural landscapes still have a relatively unchanged structure, which serves as one of the last indicators of historical practices and land organisation.

The Mediterranean climate is suitable for agriculture, but its erratic rainfall along with the island's location in the karst environment, which is characterised by high erosivity, complexity of landforms, scarce soil-limited mainly to the karst valleys and sinkholes [32] (pp. 102–103), and lack of surface water [33,34], has caused human agricultural intervention to be strenuous and well thought-out. In this sense, the dry stone wall enclosures and terraces today, along with their multifunctional role and importance in the postmodern age [35], became a symbol of sustainability in agricultural landscapes. In conjunction with the dynamics of public interest and civic activism, growing touristic popularity of the Croatian coast, and increasing institutional acknowledgement, Croatian dry stone wall landscapes became a part of a "sought-after commodity which takes place in a national shop of non-inventoried items" [36] (p. 464).

The Cres olive grove is representative of such landscapes, having certain distinguishing features. It is located in a semi-flooded, differential erosion-formed karst valley with dolomite and anthropogenic soils [32] (p. 32) surrounding the town of Cres (Figure 1). Among 69 registered outstanding agricultural landscapes of Adriatic Croatia, the Cres site was one of only two examples of agropastoral polyculture and one of the few that have preserved historical morphological relations to urban fabric [37] (p. 129). Today, it is one of the largest homogenous traditional olive grove areas in the Croatian Adriatic.

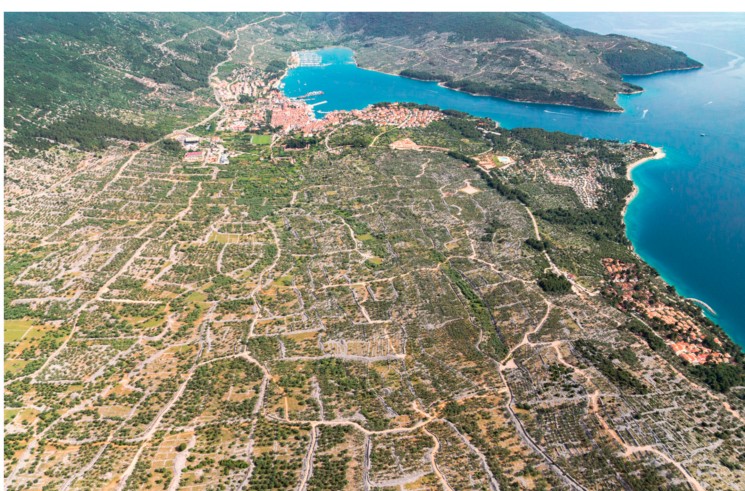

**Figure 1.** Olive groves and the amphitheatrical karst valley surrounding the town and bay of Cres (Goran Andlar, 2019).

The research area was defined on the basis of a discernible dense network of dry stone walls, as recognized from the aerial imagery from 1953 [38], thus encompassing the full

extent of the historically cultivated area (21 km$^2$), including currently overgrown parcels within the Cres karst valley and eastern slopes of the outer bays (Figure 2). Today, the area accommodates approximately 250,000 olive trees and 1500 sheep, amounting to 90% and 10% of the island's total, respectively [39] (p. 23).

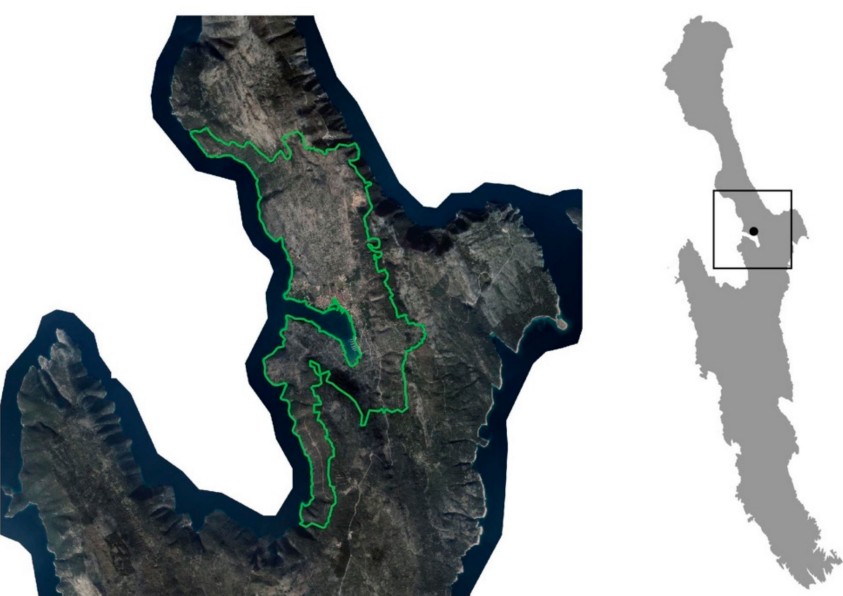

**Figure 2.** The research area and its location within the island of Cres. Basemap: digital orthophoto 2014/2016 (DOF 2014/16 WMS) [40].

Approximately 2000 inhabitants live in the town of Cres, with only a small number of professional farmers [39], as agriculture in Cres is today an economic activity ancillary to tourism. Historically, this area was representative of Mediterranean polyculture in its immediate surroundings, with a significant prevalence of market-oriented olives and vines. After the vine disease at the turn of the 20th century, it became, generally, an olive grove monoculture area, and after the 1950s, an agropastoral area. The multifaceted blending of the legible dry stone walls and other dry stone structures and the area's extensiveness and current active use represent a rarity in the national landscapes, and a valuable land-use example of the Mediterranean cultural landscapes, where abandonment, neglect and illegibility of dry stone walls is increasingly pronounced. The Cres olive grove landscape therefore became a part of a relevant global environmental and cultural and sustainable development discussion. However, it is still neither adequately researched nor appropriately managed, which, together with the demographic decline, had an additional negative effect on spatial trends: marginal parts of the area are visibly deteriorating under reforestation caused by agricultural abandonment, and immediate surroundings are pressured by the expansion of touristic and housing infrastructure. Increased reliance on tourism enables higher profitability of certain products, but it simultaneously discourages agricultural activity, accelerates abandoning stone masonry, and deprives this agricultural system of its main elements—people and practices deserving for its upkeep.

To date, the Cres olive grove has been addressed in various valuable works, but only partially [32,41–46]. Notable joint non-academic achievements by Josip Kremenić and Tarcisio Bommarco paved the way to this topic. Their work has been published in the form of an internet blog and conference proceedings [47,48] or as unpublished material. Kremenić's work was later revised and published posthumously in local conference proceedings [49]. Apart from the aforementioned, there have been no studies which capture this landscape as a multifunctional whole or which acknowledges its relevance within the current multidisciplinary discussions. Based on the authors' assumption that the Cres olive grove is an important conserved cultural landscape in the national and international perspective,

and as such not adequately explored or protected [37,50], the aim of this work is to clarify past and current circumstances surrounding and affecting it. In this paper, a preserved or conserved landscape is considered a landscape which maintains, for the most part, its main historical structural composition (relief and dry stone wall patterns and structures), traditional knowledge (agricultural practices and locally used terminology) and production of its traditional crop; nonetheless, there have been visible spatial changes, reduction of traditional practices, and newly included (pastoral and touristic) economy. Therefore, it is a landscape that aptly adjusts to external impacts but maintains the main constituents that define it as a cultural multi-value landscape. This study focuses on two main pillars of landscape preservation: (1) dry stone wall structures and (2) the agro-pastoral practices, giving an overview of its historical formation and current management and trends. The goal is to establish knowledge that can be used as a foundation for the management of this area and present as a good practice example for the preservation of historical landscapes in the Mediterranean region.

## 2. Materials and Methods

In order to assess the current and historical character, as well as the trends and conditions of the landscape and dry stone walls, an integral approach was applied, synthesising natural and human tangible and intangible factors (geomorphology, geology, land use, dry stone structures, economic history, agricultural practices, and local terminology). It involved desk- and fieldwork. Desk research included historical, ethnological, geographical sources and GIS data inventory and analysis. In order to analyse landscape features, history and agricultural practices, planning, protection, and management, diverse sources were reviewed: site-specific historical and ethnographic articles and studies (conference proceedings, journal articles, monastery archives, statute, and unpublished manuscripts); local development strategies, Cres olive oil Protected Designation of Origin documentation [51] and Town of Cres spatial documentation [52] for past and current agricultural practices, management and protection; Croatian and international articles and studies on dry stone wall shaped historical landscapes in order to place the case study within Adriatic and Mediterranean context.

Deskwork further included the establishment of a GIS database. Based on the available cartographic data, it includes: Austrian Empire cadastral maps from 1821 and its rectifications from 1837 and 1847 [53], aerial imagery from 1953, 2014, 2016, and 2020, Croatian Base Map 1:5000 available from the State Geodetic Administration geoportal [40] and ARKOD land use registry [54]. These data enabled creation of an inventory (quantification, distribution, and locating) of spatial features (dry stone walls, pathways, and land use) as well as further analysis, such as evaluation of the interaction of various features, temporal land use analysis (from 18th century onwards), and dry stone wall pattern classification.

Field research was first conducted in 2014 as part of a different project [50], and afterwards on multiple occasions in 2019, 2020 and 2021. It involved detailed inventory and analysis of particular stone structures, assessment of their characteristics, and condition and current land use tendencies. It was supported by terrestrial and aerial (drone and airplane) photos and establishment of a geotagged photo database. Research was supplemented and directed by formal conversations with local agriculturists (See: Acknowledgements), conducted in order to gather knowledge on the terminology and location of dry stone wall structures and agricultural practices.

## 3. Results

### 3.1. Historical Formation of the Dry Stone Walled Landscape of the Cres Olive Grove

The town of Cres is an ancient town [55], a Roman Empire municipium, known since the 1st century as Crexa/Crepsa, and a constantly inhabited settlement since. It became the Cres-Lošinj archipelago's administrative centre in the 15th century. Based on the numbers of amphorae that were exported from Istria and the Kvarner islands, mainly to the Po Valley [56], it is known that Cres was a centre of oil and wine production in

Roman times. The Roman rule created the basis on which the Cres economy relying on olive and vine growing would develop in later times. The absence of a regular grid of the ancient Crexa is consistent with the assumption of advanced deurbanization of the ancient settlement during the early Middle Ages when Osor, the seat of the diocese, was the only one to preserve the functional and formal features of a city [57] (p. 20). However, it is assumed that a major part of the dry stone wall enclosing and cultivation of what we know today as the Cres olive grove happened in modern history (Figure 3) between the 16th and 19th century, under the Venetian and Austrian rule, mostly driven by the production of olive oil and wine, respectively, and the increased economic prosperity during the Austro-Hungarian rule (1814–1914). The hypothesis that the most intense transformation of the Cres environs was olive oil driven and happened mostly between 16th and 19th centuries (See: [49]) is based on available historical sources and notions. In them, there is little mention of olive oil production before the 1500s [58] and an almost full reclamation of the area by 1821, which can be observed on Austrian cadastral maps and detailed land measurement from 1900, in which Cavallini reported about 1288 ha of olive groves and 300,000 olive trees [59] (p. 37). Furthermore, an increasing number of oil processing facilities can be traced to between 1600 and 1900 [60] (p. 107). In 1770, Alberto Fortis admired the people of Cres who "shaped the surroundings of their town into a garden to a considerable height" [61] (p. 141), stated that the work of clearing the stones was performed continuously; that the Cres olive oil was considered to be the oil of the highest quality in the countries of the Venetian Republic and the richest product on the island (pp. 59–60) and added: *"We should be sending to the island, but mostly to the town of Cres surroundings, sons of our arrogant doctors and laggards who inhabit our tamed and gentle hills of the terraferma. It deserves to be their Athens and an example, if for nothing else, then only to convince them that there does not exist such ungrateful soil that craftsmanship, diligence and care cannot transform, with exceptional results."* (p. 39).

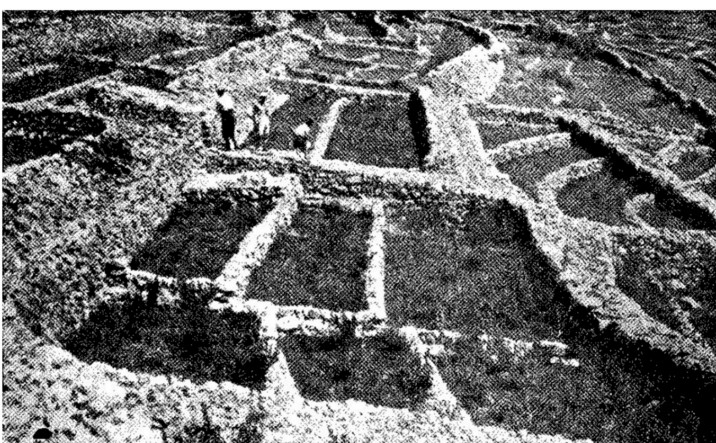

**Figure 3.** Well-maintained dry stone terraces surrounding the town of Cres in the late first half of the 20th century (Franjo Jardas [62]).

Since 2006, Kale has been arguing that the most prominent Croatian dry stone wall landscapes (agricultural terraces and dense orthogonal raster shapes) have been triggered by the same globalization that is pressuring them today; that these landscapes are recent (within the last 150 years) and were formed as a result of short-lived, and thus intense, transformations of the environment, as part of the large-scale events that the Adriatic region was part of [36]. These events are primarily related to an increase of prosperity and population in general and very favourable conditions in viticulture. Similar conclusions have been formed for the major part of the Italian terraced landscapes [63] (pp. 9–14). However, by GIS vectorisation of the cadastral maps from 1821 it was measured that 80% (17.5 km$^2$) of the research area was already agriculturally active (Mediterranean triad), with a significant prevalence of olive groves: almost 50% of the site were olive grove

monoculture cultivations, with olive trees present in 2 additional km$^2$ of arable land, pastures and vineyards. Pastures (20%) and vineyards (20%) are the next most represented types of land use, distributed in more remote areas (Figure 4).

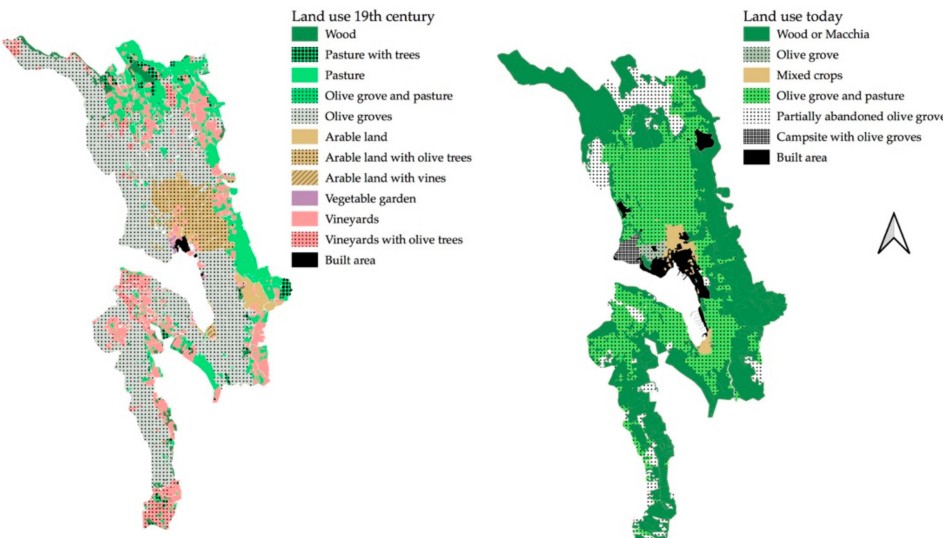

**Figure 4.** Land use in 1821–1837 and 2016. GIS elaboration based on cartographic analysis of Austrian cadastral maps [53], digital orthophoto imagery from 2016 [40] and ARKOD land registry [54].

Remoteness from the city and tessellation with forest patches may suggest the beginning of the rise of viticulture and the later agricultural acquisition of less suitable lands visible on the orthopho aerial imagery from 1953. Therefore, the landscape of the town of Cres is an example of an older land transformation, guided largely by olive oil production. Further land transformation (from agriculturally inactive to terraced vines), governed by the ephemerally favourable conditions in viticulture, which occurred in this region between 1870 and 1890, is legible as dense orthogonal disposition of the dry stone walls and is visible in the neighbouring area called *Lovreški*. Within the Cres valley, these circumstances caused the local farmers to partially neglect existing olive plantations by intensely cultivating vines wherever available, either by pulling out olive trees or planting within the existing olive groves. Grapevine disease, which destroyed the island vineyards from 1890, left behind either empty or overgrown dry stone wall parcels and, close to them, dry stone or concrete water reservoirs as a land use indicator (locally: *vaške*) (Figures 5 and 6).

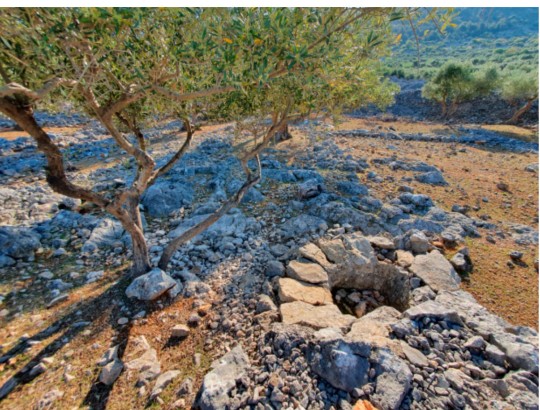

**Figure 5.** *Vaške* (from Italian: *vasche*) as land use indicators, normally where the vines were present Constructed by the use of both concrete and dry stones (Tanja Kremenić, 2020).

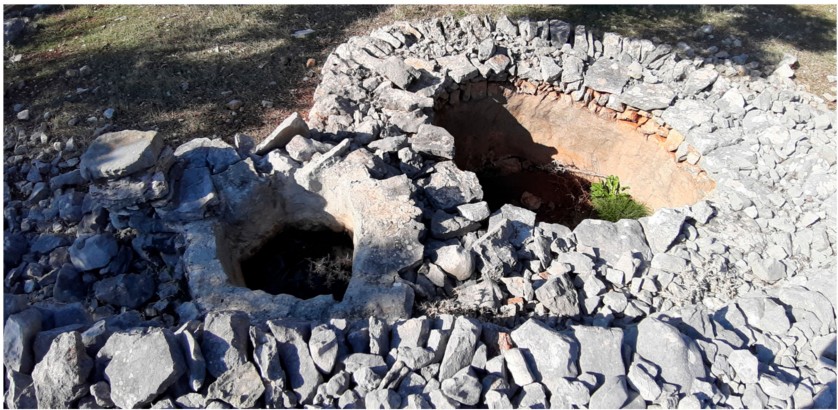

**Figure 6.** *Vaške* appear mainly in pairs: one as a water reservoir, the other as an pesticide reservoir. Several still retain blue colour traces from copper (II) sulphate (Karla Glavočić, 2019).

### *3.2. 1950s—Beginning of the Agro-Pastoral Organisation and Technological Advancement*

Sheep breeding and olive growing have always been two of the main agricultural activities on the island of Cres [32] (p. 197), but until the 20th century, they existed separately [51]. Although introducing sheep to assist with olive growing is not an uncommon agricultural practice, it is rare in the Croatian Adriatic. Furthermore, in the example of Cres, it was introduced as a full bottom-up approach: after observing the benefits of the accidental releasing of sheep into the olive groves (sheep's foraging was preventing the olive grove from overgrowing with natural vegetation, without damaging the trees or the yield) in a more distant area in the northern part of the island (*Bok*) in the 1920s, members of the Agricultural Cooperative decided to apply this "olive-sheep" model around the town of Cres in 1956 [51]. Securing additional "non-human" labour was necessary in order to save the olive cultivation after the demographic decline in the early 20th century, and especially after an exodus after WWII (y. 1910: inh. 4196, y. 1945: inh. 3087, y. 1953: inh. 1670) [32] (p. 309).

A systematic enclosing of the olive grove was completed prior to sheep introduction, primarily by wire fencing, but also by constructing higher dry stone walls and a multicellular sheepfold for their gathering and separation (Figures 7 and 8).

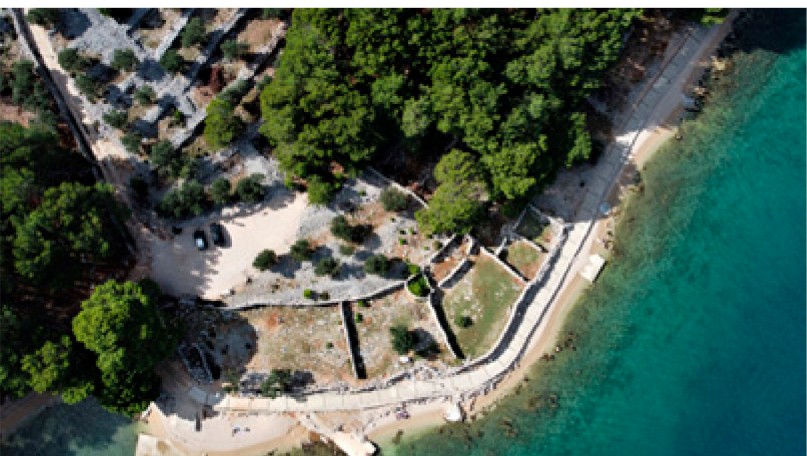

**Figure 7.** Multicellular sheepfold (*mergar*) within the olive grove (Dražica bay). It is no longer in use (Tanja Kremenić, 2020).

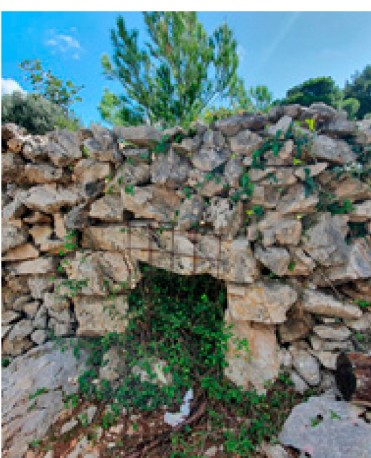

**Figure 8.** A sheep passage (*škuja*) from one of the "cells" of the multicellular sheepfold (Tanja Kremenić, 2020).

Considering that agropastoralism was a novel approach, not all of the Cres olive growers were eager to become involved in it, therefore the Cooperative had to organise enclosing of even smaller private parcels, which is why higher dry stone walls inside of the olive grove are still visible. Today, sheep graze freely across the whole area (Figures 9 and 10).

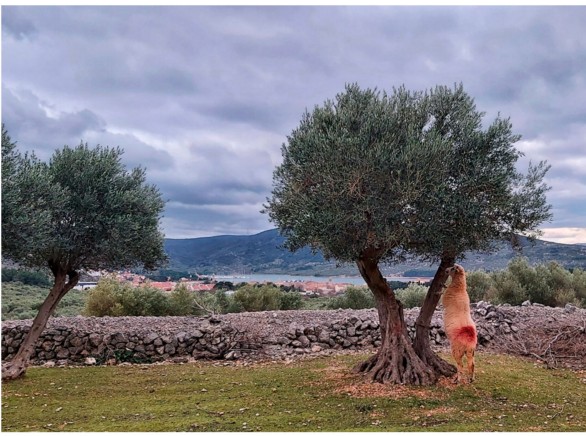

**Figure 9.** Sheep's foraging is preventing the olive grove from overgrowing with natural vegetation, and, since the trees are old and developed into a certain height, it does not affect the yield (Dubravko Matić, 2021).

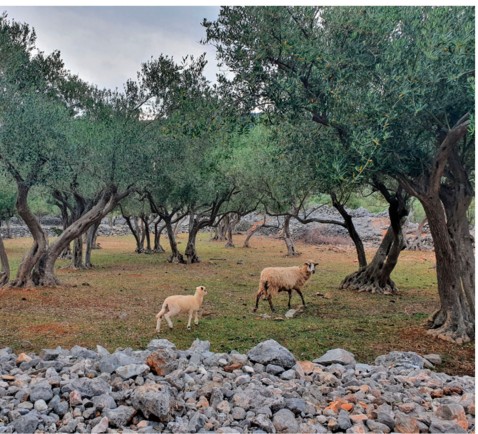

**Figure 10.** A common sight within the Cres olive grove: the dry stone wall-olive-sheep association (Tanja Kremenić, 2020).

In the 1980s, the Cres olive grove was included in an FAO project, and as such was the only one on the Croatian Adriatic coast [46] (p. 36), [64] (p. 105). Encouraged by this external recognition and growing demand for olive oil, Cres farmers started renewing the olive groves. In 1990, there were 543 ha of active olive groves, about 30% of which consisted of dry stone walls and piles account, according to visual estimates. Therefore, the cultivated olive groves comprised only 380 ha [65] (p. 92), testifying to a 60% reduction from its peak year (1900). Today, Ulika Association reports that the Cres olive grove counts approximately 600 ha, 130,000 active olive trees and 122 members [66].

### 3.3. Past and Current Agricultural Practices in Olive Growing

Cres olive oil was the most important product of the island. According to Fortis, it was of the highest quality in the whole Venetian Republic and the "art of pruning was nearly perfected, while on the Apennine Peninsula it was almost unknown". The canopy was kept low, probably due to high exposure to the strong winds of bora. Processing was performed by a slightly modified Tuscan method. The harvesting method also contributed to the quality—unlike in certain parts of Italy where olive growers struck the branches with sticks, damaging both the branches and the fruits when they fell to the ground. In Cres, the fruit was picked by hand directly from the tree, thus avoiding fermentation and rotting [61] (p. 43).

Today, the cultivation model is still predominantly traditional and low-density (230 trees/ha) [51], extensive and of low environmental impact, with the adoption of modern agro-technical methods. All the olive trees are located on the dry stone wall enclosures and terraces and currently disallow further mechanisation. The rejuvenation of olives by pruning is the prevalent and most effective agro-technical measure, considering the old age of the olive trees (there are almost no new trees planted in the last 60 years). Introduction of sheep preserved the structural form of this landscape on environmentally apt terms: overgrowth is prevented without the use of herbicides and without the need to use mechanization that would demolish the dry stone enclosures and terraces. Furthermore, it contributed to moisture retention, modest fertilization, and browsing of foliage left by pruning, leaving the branches available as firewood. It entails disadvantages as well, such as difficulty of restoration from tree shoots and inability to lower the tree canopy [64] (pp. 111, 114). Another important agro-technical measure is protection against pests (*Rhodocyrtus cribripennis*, *Bactrocera oleae*, *Prays oleae*), as pests are the main cause of low yield. Harvesting and pruning are performed simultaneously and manually (Figure 11), one plant at a time, with the help of the branch shakers (Figure 12). Harvested olives are taken directly to the oil mill in the town of Cres, where they are processed within 24 h. Until the 2000s, it was still a common practice to keep olives in sea water between harvest and processing. Although it was already praised in the late 18th century, today's olive oil from Cres differs from the oil produced as late as 30 years ago. In 1975, the Agricultural cooperative bought the first oil mill with a centrifugal extractor in the Republic of Croatia.

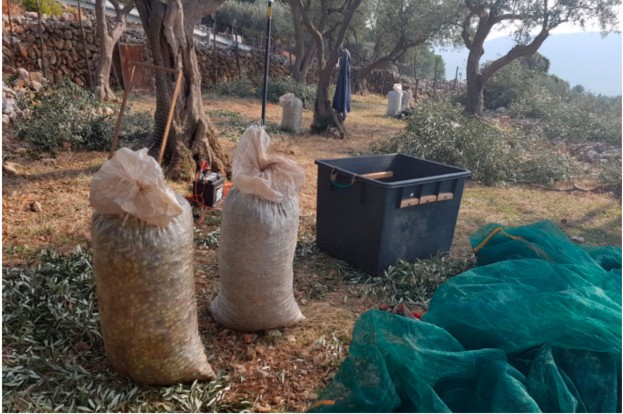

**Figure 11.** Olive harvesting is still performed mostly the traditional way (Tanja Kremenić, 2019).

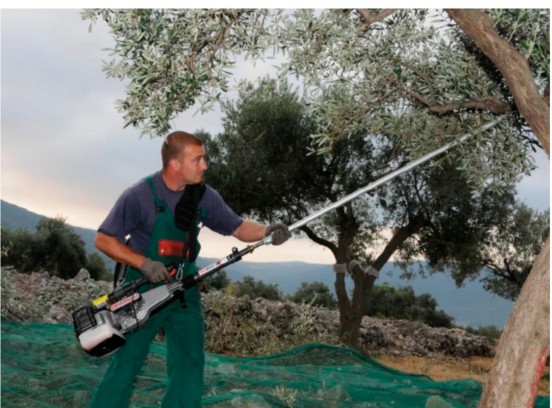

**Figure 12.** The branch shakers are the only mechanical addition—further mechanisation is not implemented since all of the olive trees are located on the dry stone wall enclosures and terraces [39].

### 3.4. Cres Olive Grove Today-Agro-Pastoral Touristic Multifunctionality, Compromises and Protection

### 3.4.1. Current Land Use

With the introduction of sheep, the Cres olive grove moulded into an agro-pastoral site, and with increasing tourist demand in recent years, an agro–pastoral–tourist site, having multiple effects on the landscape. The symbiosis of tourism and olive groves is manifested in the town's camping site. Camping plots are organized on the existing dry stone wall terraces, where a large number of trees is still used for harvesting. Widening of the historical roads for vehicles since the 2000s, which was directed by the Cres farmers gathered in Ulika Association (Figure 13), besides providing easier access and recovery of the abandoned olive groves, created recreational pathways used by both locals and tourists (Figure 14).

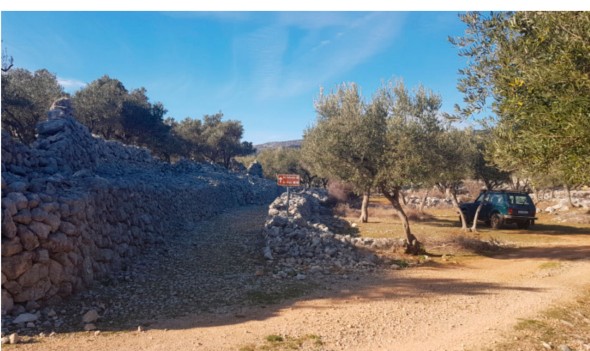

**Figure 13.** One of the most noticeable recent landscape changes in the olive grove are the new roads for vehicles, which sometimes imply perforations of the historical paths. No compensatory measures are currently anticipated (Tanja Kremenić, 2019).

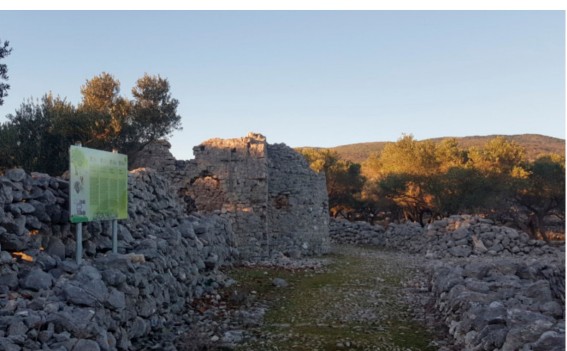

**Figure 14.** As the area has become a touristic site, recent landscape changes also included the addition of educational signs (Tanja Kremenić, 2019).

The construction of these roads entails dry stone wall crushing, after which no compensatory measures are anticipated. In the last decade, the immediate town surroundings have rapidly been transforming into a space of new uniform housing buildings, which is currently the most visible and pervasive form of Cres olive grove landscape intervention. Apart from consuming agricultural terraces, it interrupted certain historical paths that led to distant parts of the olive grove and have been visual and tangible symbols of the inseparability of the agricultural background and town centre (Figures 15 and 16).

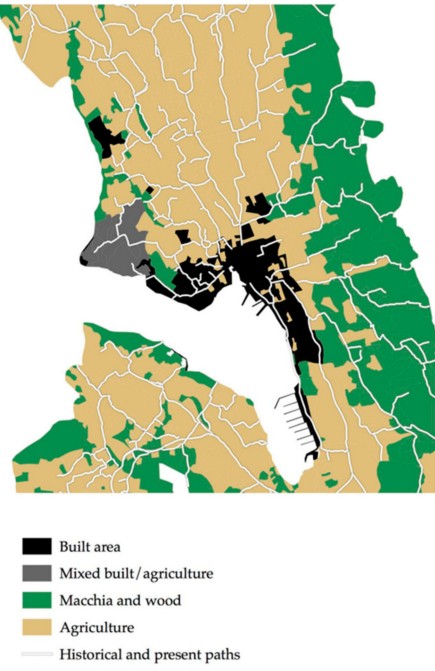

**Figure 15.** The structural interconnection of the town of Cres with its agricultural zone is noted in the network of dry stone wall lined pathways.

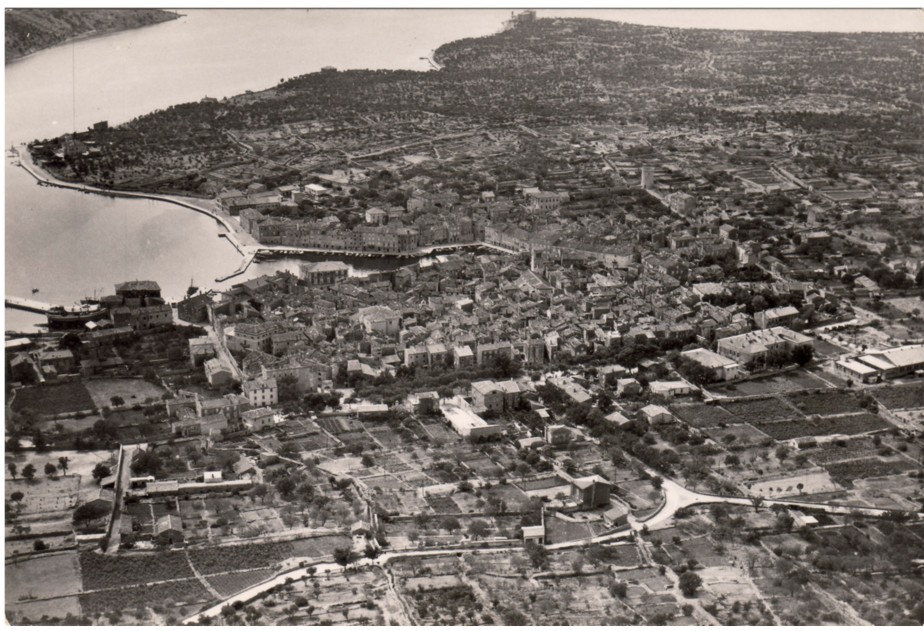

**Figure 16.** One of the rare high-quality photographs taken in mid-twentieth century. Besides the paths that lead from town to olive grove parcels, an arable crop land close to the town is discernible, which is today mostly ground for new housing construction (D. Rendulić—Agencija za fotodokumentaciju Zagreb, 1960s).

GIS analysis of aerial imagery shows that forest and macchia comprise the dominant land cover (54%) of the research area. Active olive groves still occupy a large share (46%), including 9% of partially abandoned olive groves. Most of the olive trees are still suitable for revitalization. Polyculture in arable land occupies less than 1% (Figure 4), mostly in the town's southern area with alluvial soils (*Piskel*) and in the immediate town surroundings.

3.4.2. Legal Protection and Management

This area, with its associated values, is a subject of protection within the following regulations:

- Forty-five percent (9.78 km$^2$) of the historically cultivated area, including the town area, has been protected within the Town of Cres Spatial Planning documents as "especially valuable cultivated landscape", due to its agricultural landscape qualities. As such, it has been intended exclusively for agricultural production [52] (p. 6843);
- Thirty-five percent (7.34 km$^2$) of the historical olive grove area is distinguished as P2 ("valuable arable land"), in which any construction that is not in the function of agricultural activities is prohibited. However, it has been protected due to its "cultural landscape values" [52] (p. 6849), which is an exception in Croatian spatial planning, since P2 category is normally based on soil quality;
- Cres extra virgin olive oil is the first Croatian olive oil that received the PDO quality mark in 2014, following the acceptance of the application documentation written by the local Ulika association. This designation applies to all the extra virgin olive oil made from the autochthonous *Slivnjača* and *Plominka* varieties, which naturally meet all specified quality parameters and production sequences, considering the entire island of Cres as a geographical determinant, but without addressing the historical landscape as a prerequisite for such yield. Seven years later, only three producers adopted the quality mark. This is not only representative of the island's circumstances, but of the lengthy application procedures and the costs of inspection and verification in Croatia [67] (p. 68);
- With the protection of "art of the dry stone walling" as an immaterial good by the Croatian Ministry of Culture in 2016, a list of craftsmen in dry stone walling on national level has been established. Although the art has been protected on the level of UNESCO in 2018 as well, and nonetheless many of the olive growers practice the necessary renewal of the olive groves by dry stone walling, few applied to be included. From 35 natural persons and 18 legal persons in Croatia, only 3 private persons and 1 legal person from the island of Cres were included, all from the village of Orlec [68].

*3.5. Typology of Dry Stone Structures and Patterns*

GIS vectorisation revealed more than 2605 km of dry stone walls on an area of 21.6 km$^2$ in 1953 (Figure 17) with a high density of 120 km/km$^2$ (or 1206.8 m/ha), which can be expressed as 1 dry stone wall every 8 m. This density is the highest in all of the dry stone wall landscape of the archipelago. This number will eventually be reduced by more than 150 km of dry stone walls (6%), as they are situated in a construction-intended zone and in the way of perforated gravel roads.

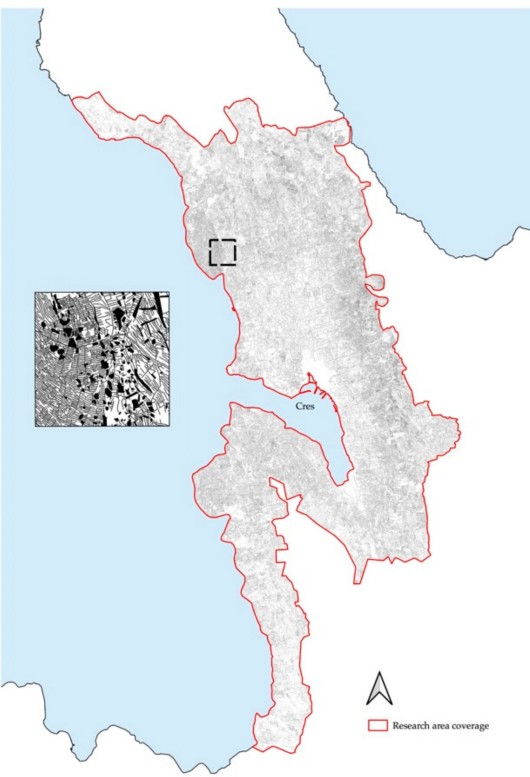

**Figure 17.** GIS vectorisation of all of the dry stone walls within the Cres olive grove research area.

3.5.1. Typology of Dry Stone Structures

In this study, dry stone structures are categorised following a dry stone inventory model proposed by Šrajer [14] (pp. 143–146). The words in italics present local names for the structures in nominative singular form.

1.  Linear structures

    - Double walls (*duplica*) are the predominant ones. They vary in height (0.3–2 m), adapting to the terrain configuration, forming dry stone wall retaining terraces. Locally, the term used is *barbakan*, which has several meanings. It denotes the retaining dry stone wall, a low dry stone wall (0.3 m), but also the entire terrace step, with soil included (Figure 18);
    - Single walls (*unjulica*)
    - Dry stone walls of combined construction have the first half as a double wall composed of smaller stones, followed by a single wall with larger ones (Figure 20);
    - Dry stone walls of greater height, topped with capstones, which are diagonally positioned (*ozubi*). When their function is one of dividing a pasture into smaller parcels to facilitate sheep gathering, they are called *tres;*
    - Dry stone water channels (*konal*, *konalić*) (**??** and Figure 21);
    - Dry stone wall linedd paths (*klanec*, *klančić*), sometimes paved with smaller pebble stones. Many are overgrown in vegetation, used as material disposal or consumed by new roads. Based on available digital orthophotos, more than 100 km of dry stone wall lined pathways have been measured in GIS (Figure 15).

2.  Individual buildings

    - Dry stone wall shelters (*kućica*) are located within almost every parcel, usually closer to the town, and are predominantly constructed in a rectangular form with a roof made from tiles and wood. Shelters with a circular ground plan with a dry stone corbelled dome can be found in significantly fewer numbers in more distant and overgrown areas (Figures 22–25). Their rare occurrence is particular since they are the norm in the neighbouring regions of the Croatian Adriatic.

3. Individual structures

- Dry stone piles (*menik*, *menicić*) are elongated, asymmetric or symmetrical with or without a retaining dry stone wall, often with an additional dry stone wall built upon them (Figure 26);
- Multi-cellular dry stone wall sheepfold (*mergar*) (Figure 7) is built of single-type walls or combined ones, with specific pastoral details such as a sheep passage (*škuja*) (Figure 8);
- Similar to a shelter is a dry stone "oven" (*pećnica* or *fornaž*), which is used for drying figs, in case they have been affected by rain (Figure 27);
- Masonry ponds (*lokva*) are today mostly built from concrete. The ones constructed in dry stone are rare and are located in more distant areas.

There are several other architectural elements, such as shoulder-high double walls intended for rest (*pocivalići*), holes intended for lowering the height at which the bags for collecting olives or grapes were placed and stairs that functioned as avenues for accessing the groves and for connecting terraces (Figures 28 and 29).

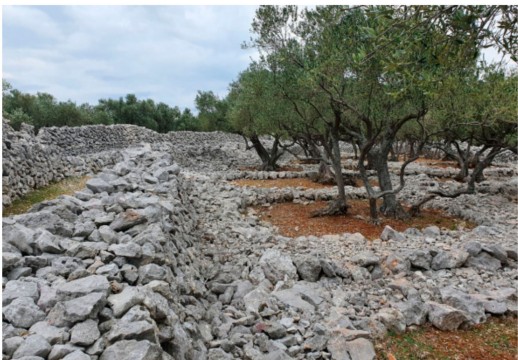

**Figure 18.** From left to right: a dry stone wall pathway, higher double wall (*gromača*), lower dry stone walls (*barbakan*) (Tanja Kremenić, 2020).

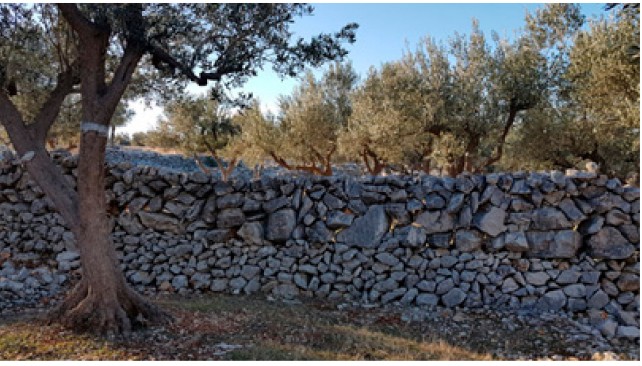

**Figure 19.** Island of Cres specificity—a dry stone wall of combined construction, composed of both a double and single dry stone wall (Tanja Kremenić, 2020).

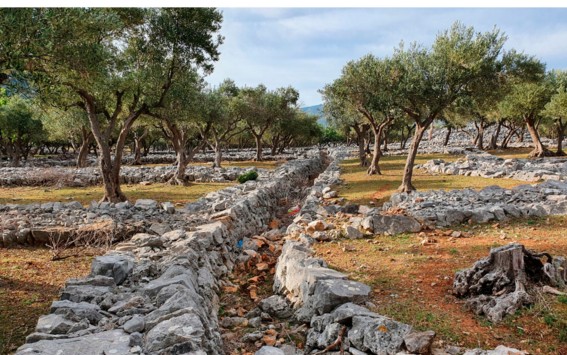

**Figure 20.** The drainage system within the Cres olive grove consists of dry stone channels which gather the water from the terraces, thus avoiding the rainfall to damage the yield (Tanja Kremenić, 2020).

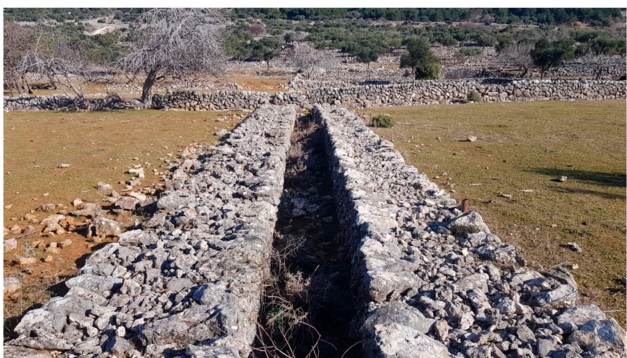

**Figure 21.** The dry stone channels serve to direct the water from the heavy rainfalls and flooding (Tanja Kremenić, 2020).

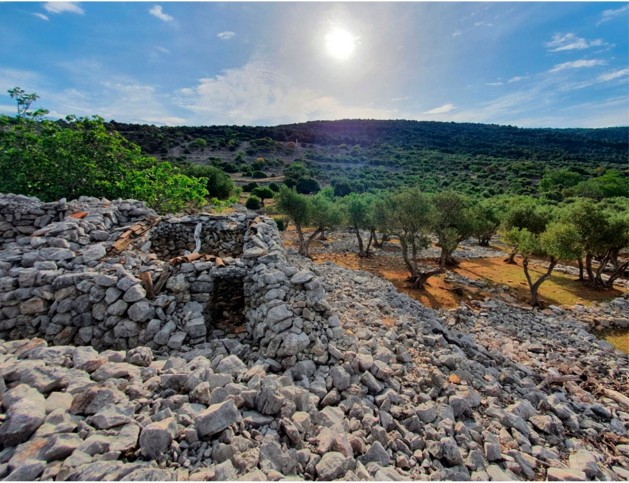

**Figure 22.** Dry stone wall shelters are for the most part rectangular and constructed on dry stone piles (Tanja Kremenić, 2020).

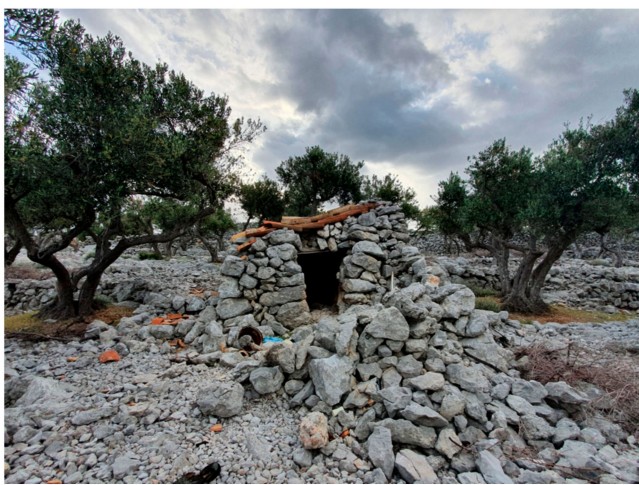

**Figure 23.** The typical dry stone wall shelter, with a maintained single pitch roof made from tiles, is in the Cres olive grove (Tanja Kremenić, 2020).

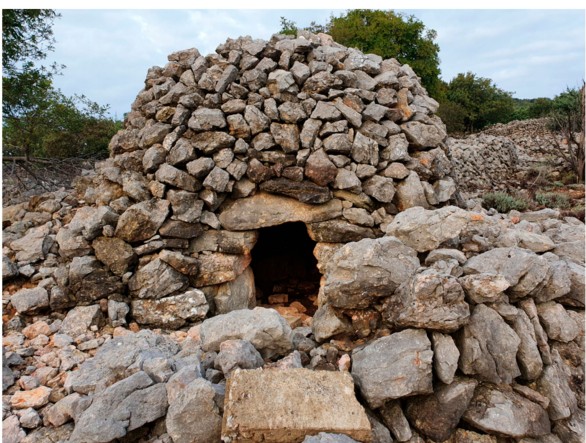

**Figure 24.** Dry stone shelters with round plans and stone roofing are considerably less frequent and less known on the islands of Cres and Lošinj in comparison to the rest of the Croatian coast (Tanja Kremenić, 2020).

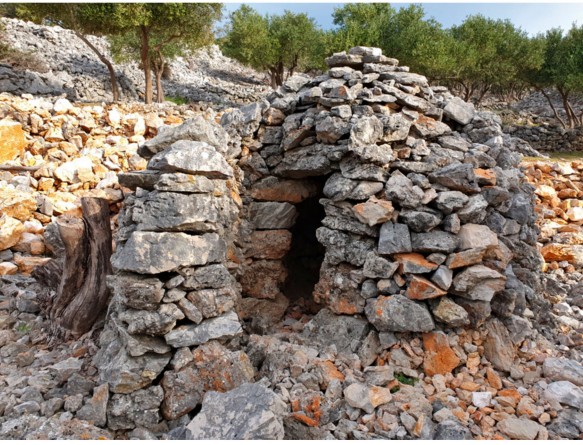

**Figure 25.** Besides being rare, dry stone shelters with round plans and stone roofing differ among themselves (Tanja Kremenić, 2020).

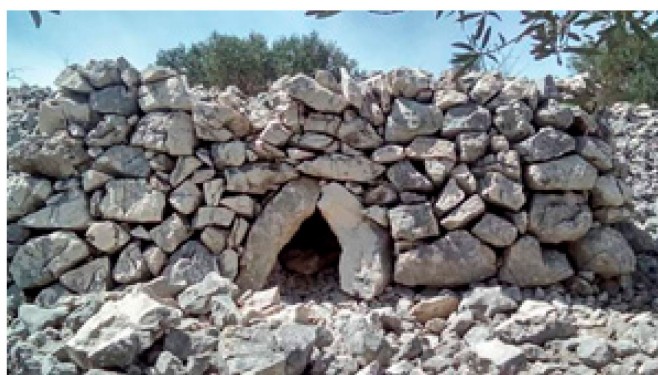

**Figure 26.** A stone pile with a dry stone wall on top (Goran Andlar, 2016).

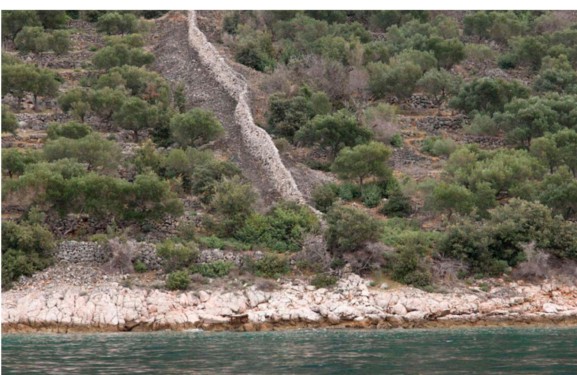

**Figure 27.** Pećnica is a dry stone wall structure, similar to dry stone wall shelters, which is used for drying figs in case they have been affected by summer rain (Loredana Velčić, 2020).

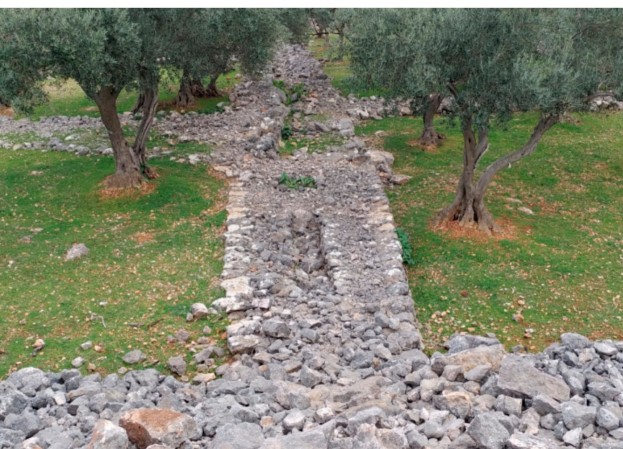

**Figure 28.** Dry stone details: a hole intended for lowering the donkey's height on which the bag with olives or grapes was placed (Franko Fučić, 2020).

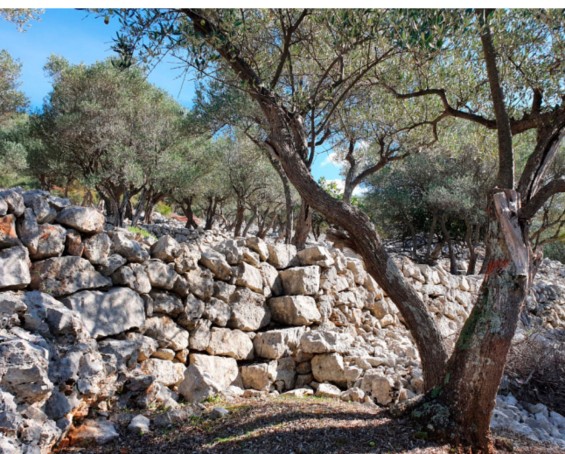

**Figure 29.** Dry stone wall details: stairs as main avenues of approach to the olive grove and within terraces (Tanja Kremenić, 2020).

### 3.5.2. Dry Stone Wall Landscape Patterns

A dry stone wall landscape pattern is here considered as a spatial arrangement of dry stone walls in a floor plan view. Part of the Cres olive grove follows a particular spatial design, with the base and consecutive units of *graja* (Figure 30). *Graja* is a parcel enclosed on the sides with elongated dry stone piles (*menici*), intersected with terraces and retaining walls (*barbakani*), and interconnected with a network of dry stone wall lined pathways (*klanci*). Originally, one family had the right to transform pastoral or agriculturally inactive land into an agricultural plot. Upon reclaiming one plot, they had the right to request another one [49]. The *graje* form a large share of the Cres olive grove, except for the bay slopes, which are characterized by dense regular terraces.

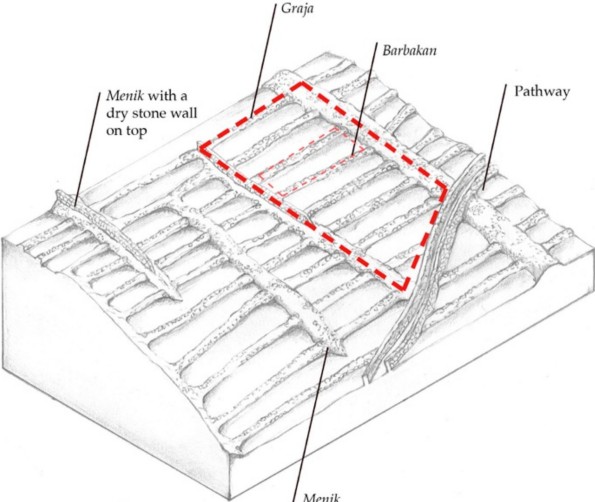

**Figure 30.** Main components of an agricultural plot within the Cres olive grove named *graja* (sketch author: Anita Trojanović).

*Graje* are not the only agricultural unit. The identification of other dry stone wall landscape patterns involved interpretation of various natural and cultural aspects: geomorphology, soil, agricultural practices, local dry stone wall typology, historical planning system and 2D and visual interpretation. Their classification relies on the authors' earlier work [7,8]. Therefore, this typology starts from main types—in this case, crop karst enclosures and terraces—while further subtypes are distinguished on a structural basis.

Crop enclosures are defined as dry stone wall enclosed plots found in rocky karst areas with scarce or absent soil, where cultivation was determined by stone removal and

its stacking in walls and piles, thus creating small, fragmented patches of soil for crop growing. In this area they cover flattened or mildly sloped areas where terracing was not present.

- Figure 31a: regular enclosures with variations of regular patterns of inner stone structure;
- Figure 31b: irregular enclosures with regular patterns of inner walls and retaining walls;
- Figure 31c: irregular to regular enclosures with massive enclosing and inner piles;
- Figure 31d: irregular enclosures with irregular patterns of inner stone walls and piles.

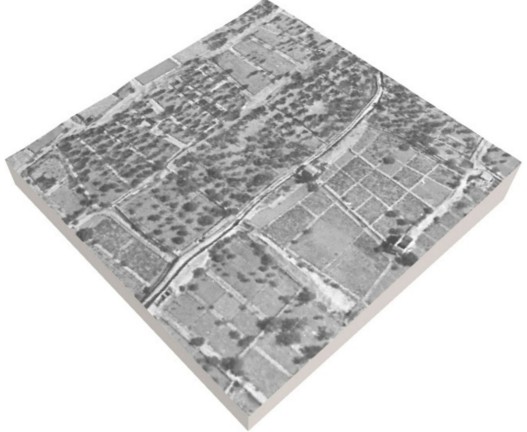

(**a**): regular enclosures with variations of regular patterns of inner stone structure

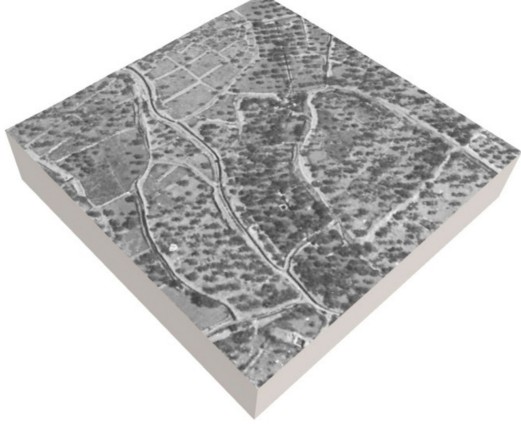

(**b**): irregular enclosures with regular patterns of inner walls and retaining walls

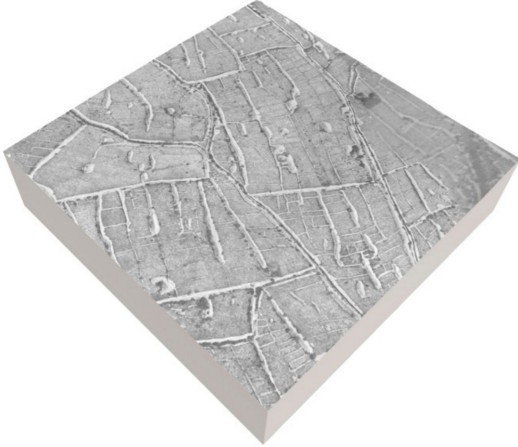

(**c**): irregular to regular enclosures with massive enclosing and inner piles

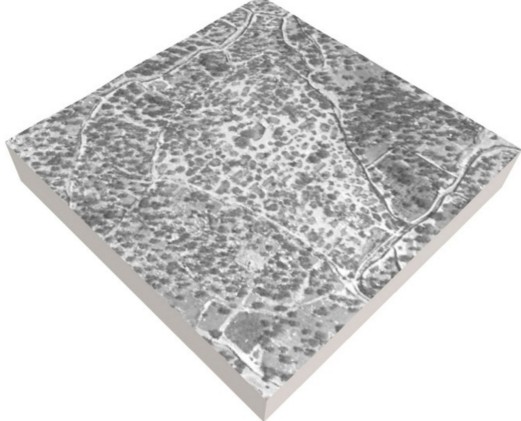

(**d**): irregular enclosures with irregular patterns of inner stone walls and piles

**Figure 31.** Types of karst enclosures found in the Cres olive grove. Elaboration is based on aerial imagery from 1953 and digital elevation model.

The first subtype is regular enclosures (Figure 31a). The enclosures are for the major part irregular, but still with a regular inner structure (Figure 31b). Only occasionally can massive enclosing and inner stone piles be found (Figure 31c). Rarely, the outer and inner structures are fully irregular (Figure 31d); probably due to a great amount of stone and more pronounced relief, the regularity of inner structure is lost. Regular patterns attributed to vine terrace reclamations are located in the marginal parts of the research area. However, the type of pattern is predominantly irregular as it depends on the landform characteristics, higher steepness of the terrain causes lesser soil availability and more

complex and irregular patterns, to which the density of the massive stone piles in this area testifies as well. Regularity of the patterns is aspired to, which is noted since it is present in most of the plain area surfaces.

Dry stone wall terraces are defined as flattened plots on inclined terrain for agricultural, silvicultural or pastoral use. Types:

- Figure 32a: step terraces regular to curved patterns and narrow terraces, with or without minor enclosing pile;
- Figure 32b: step terraces regularly enclosed with vertical enclosing stone piles;
- Figure 32c: step terraces regularly enclosed, massive vertical enclosing and inner stone piles;
- Figure 32d: step terraces irregularly enclosed with spontaneous patterns of massive stone piles.

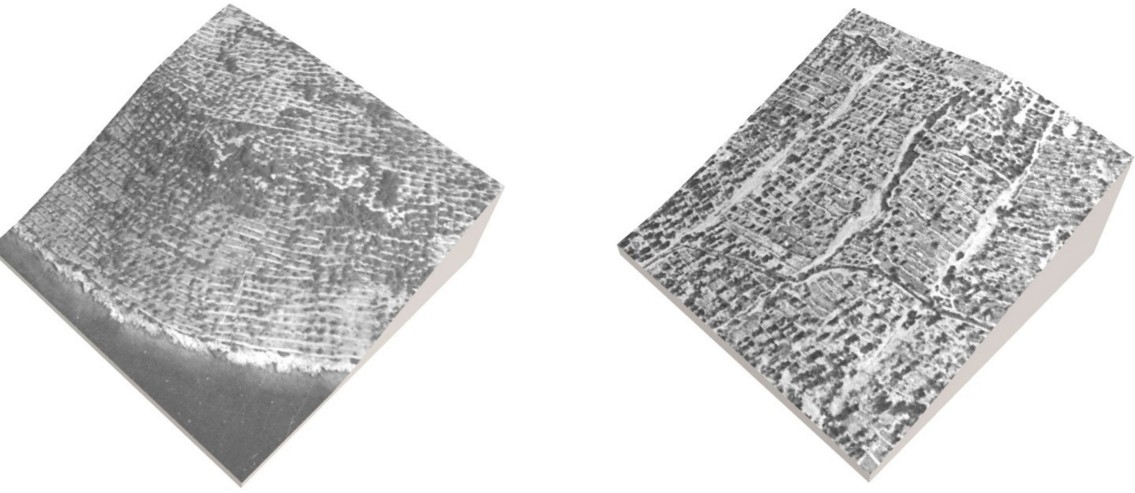

(**a**): step terraces regular to curved patterns and narrow terraces, without or with minor or without enclosing pile

(**b**): step (regular) terraces with enclosing and inner piles

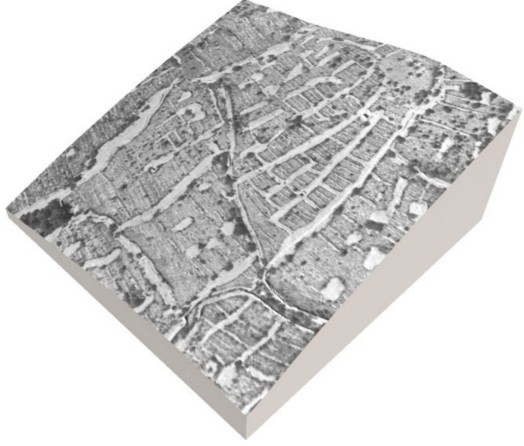

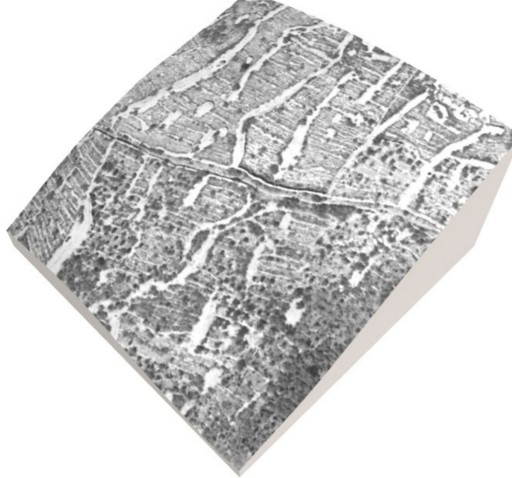

(**c**): step terraces regularly enclosed, massive vertical enclosing and inner stone piles

(**d**): step terraces irregularly enclosed with spontaneous patterns of massive enclosing and inner stone piles

**Figure 32.** Dry stone wall karst terraces representative of the Cres olive grove. Elaboration is based on aerial imagery from 1953 and digital elevation model.

Typical for this area are step (regular) terraces (Figure 32b) with stone piles following the slope and often with dry stone walls on the top of the pile. Besides massive piles enclosing the plot, inner piles are also common, resulting in type (Figure 32c). Usually, the piles are denser the higher the slope. With the complexity of the relief, the irregularity of the pile pattern grows as well, leading to type (Figure 32d). The next type are step terraces with regular to curved patterns (Figure 32a). These terraces are typical for the Croatian Adriatic in general, but less so in the Cres example.

In a series of presented patterns of both main types (1 and 2) the successive repetition of massive piles that, close up, specific modules is visible. They comprise the basic unit of this landscape (the abovementioned *graja*) and thus the local specificity.

## 4. Discussion

This paper discusses the observed values of the living historical landscape surrounding the town of Cres and the tools employed for its preservation. The value of this landscape is seen from the following aspects: age, structural and functional preservation, richness of the traditional knowledge, environmentally apt management, locally led participation in driving the changes, and functional plurality of its re-emergence as an active landscape.

The Cres olive grove is not a fully conserved relic of an (agri)-cultural traditional landscape. It would be illusory to expect a facsimile preservation from a constantly inhabited place subject to touristic economic trends, especially bearing in mind that the most extensive area was cultivated only in the short period of the greatest population increase and ephemerally favourable conditions for viticulture. The transformation of the Cres olive grove is both a locally "authored landscape" [69] and an imprint of people's personal and collective authorship, but also a result of circumstances wider than its local personality. It was guided largely by the external large-scale environmental and socio-political events in which the island was part of and therefore representative of the natural and socio-economic and political Croatian and Mediterranean context, having the vineyards and olive groves determining the spread of its dry stone wall enclosures. Its continuity was enabled by a range of adjustments. By adopting the simple upgrade in the form of an "olive-sheep" model in the 1950s, the local community introduced a novel and environmentally apt tool for the preservation of the historical landscape. Agropastoralism facilitated the maintaining of the traditional agricultural use (olive oil exploitation) and conserving the inherited capital (dry stone walls and anthropogenic soil), provided a new product (lamb), and prevented the need for additional herbicides and mechanization, which usually entails demolishing the existing enclosures and terraces. Investing in new olive oil processing technology in the 70s, the local community met the needs of the market in the form of a quality product that is today considered one of the main guarantees of maintaining the agricultural function of terraced landscapes. The perforation of new gravel roads in the last two decades reduced the area of the dry stone wall heritage but also enabled better access to certain parcels, increased oil production, and later became part of the touristic paths. Olive harvesting is important, as it compensates for the seasonality of tourism and prolongs social cohesion, thus alleviating the island's annual bipolar biorhythm [64] (p. 120). Autonomous preparation and acceptance of a PDO designation in 2014 further emphasized the importance of local participation in directing the changes affecting their territory.

The preservation of the landscape of the town of Cres was further assisted by the resistance and longevity of the dry stone wall structures. The human endeavour that had to be undertaken in order to create such dry stone wall landscapes is exceptional and difficult to understand from today's perspective. Their construction and maintenance was a common and existentially necessary secondary activity, an incidental product of socio-economic conditions, and therefore, predominantly have not been given research importance [70] (p. 11). The complexity of such interventions has only just become apparent with recently available satellite imagery, which today represents a large reservoir of information, otherwise hardly retrievable from economic history analysis. By analysing the stone wall patterns, it is possible to understand the influence of relief, hydro-geological

and climatic conditions, and crops intended, and may serve as a valuable indicator in their dating. Usually, the regularity of stone wall enclosures patterns in the Croatian Adriatic is interpreted as a result of organised and planned processes: either a Roman grid or attributed to large scale systematic land reclamations for vine growing of the late 19th century. Since most of the Cres olive grove was transformed between the 16th and 19th centuries, most of the cases of regular patterns can be attributed to the Venetian era, but it is not excluded that the immediate town surrounding agricultural orthogonal rasters are remnants of an antique grid.

Urban-agricultural integrity is still today observed in its structural and functional interconnections. Gathered yield brought to the town centre, where the oil is produced, attests to the functional connection. A structural connection is visible in a network of pathways extending in a semi-concentrical cobweb manner from the town centre (Figures 15 and 16), following the medieval central road laid in a north–south direction, which has been a part of a wider network of field roads to the surrounding olive groves and vineyards [57] (p. 31). Although these pathways to the agricultural plots have been interrupted by the concrete roads and urban expansion of low-density housing, the town's olive grove is still an integral part of the town. The use of extensive traditional agriculture related vocabulary, even if reduced, attests to the current connection between people and their agricultural area. Certain nomenclature is the same for both urban and agricultural land: word *klanec* is used for agricultural pathways, but also town streets. The term *barbakan*, used for the retaining dry stone wall, is probably of Venetian origin (*barbacane*), where it denotes an urban fortification element. However, traditional knowledge is used mainly by the elderly inhabitants and is no longer a universal phenomenon of the local community today. Alongside the lack of a detailed and accurate local dialect vocabulary, it may threaten this intangible part of heritage.

An important factor in the landscape's durability is the appeal of the island of Cres and the inclusion of the tourism industry, which brought an increased demand for local agricultural products directly to the olive grower's doorstep. It is an important variable, since the strenuous terraced olive growing practice becomes vulnerable in case of a lack of profitability. Touristic exploitation affects the broader socio-economic system [71] (p. 305), and in the example of Cres, problematic impacts are visible in questionable spatial interventions. The gravity of the issue depends on its reversibility potential; while the torn dry stone walls are restorable, tourist housing construction on valuable agricultural land is permanent. Currently, the construction of new roads has been proven beneficial for many of the olive growers and the olive oil production but it is performed with no underlying strategy nor regulated by compensatory measures, similar to the reconstruction of certain dry stone walls, which, in addition to the low number of registered craftsmen in dry stone walling and rare renewal of the dry stone walls and houses, may suggest a lower appreciation of the stone heritage within the local community.

Despite the above-described landscape values, which we believe to be of international interest, the Cres landscape has not been part of any particular protection or designation model. Its place within the spatial plan does not encompass all of its values and it has a general boundary area and vague spatial plan provisions. Due to increasing demands for areas of touristic and housing infrastructure, the coverage of this protection will probably be subject to reduction in future amendments, as tourism precedes in profitability. Within the national legislative there are two, in principle, appropriate categories for this kind of landscape. They have almost identical definitions of landscape and come from different sectors, as discussed in previous work [72]. First is "significant landscapes", within the Nature Protection Act—although having the same formal definition as the IUCN category Protected landscape/seascape V (a category designed to protect living rural landscapes whose character is based on combination of natural, cultural and aesthetic values [18]) it is usually applied only to areas of natural beauty. Second category is "cultural landscape" within the Act on the Protection and Preservation of Cultural Goods, but mostly focused on historical structures. Except that in practice these categories are very rarely applied on living rural landscapes, both lack integral and landscape scale comprehension, while

existing practices, products and multifunctionality are overlooked. On the other hand, the international GIAHS designation specifically addresses "agriculture" among its criteria, which also considers culture, traditional knowledge, food production, agrobiodiversity and landscape [73] (p. vi). These criteria have, in part, inspired the authors in the identification of the research gaps for this landscape. The importance of a global recognition of certain landscapes is beneficial, as it may prioritize scientific research.

## 5. Conclusions

This paper aimed to deliver a timely scientific intervention of a previously insufficiently addressed cultural landscape, both in terms of scientific research and management, considering: (1) the urgency of the modern era pressures (cutting ties with traditional land use practices, increased momentum of spatial interventions, vulnerability to climate change); (2) sufficiency of the preserved historical aspects (legibility of the dry stone structures, traditional practices, local community's collective memory); (3) example-setting approach of landscape preservation and its upgrade to a multifunctional land use system. Poor substantiation of relevant and reliable historical sources that will assist in the understanding of the landscape transformation before the 18th century were the main research obstacles. Therefore, the historical interpretation was hypothesised based on available archival data, and the dry stone walls and linking of their landscape pattern to historical processes. However, for their unambiguous understanding, it is necessary to conduct additional research, as the patterns mostly represent only an indicator.

Preserving the historical landscape is meaningful to the extent that it represents contemporary values, which can be identified from the viewpoint of ecosystem services. Terraced landscapes have been proven to provide a range of them [35] (p. 281). Even if these functions in the case of the Cres landscape have not yet been part of a specialized scientific research, from the review of the values of the landscape within this paper, the following can be assumed: food provision, soil erosion prevention, habitat for species, conservation of local knowledge, traditional farming and dry stone building techniques, sense of identity and sense of place, tourism, recreation and mental health, and aesthetics. It is further concluded that, currently, there is no careful management model that works to strengthen them or encompasses the values of this landscape as a multifunctional whole. Such can be achieved either by improving existing protection models or by designing new space-tailored ones. A more complete implementation of the IUCN Category Protected landscape V on a national level or FAO's GIAHS are designations in which a living agricultural landscape can aspire to.

Continuous investing in the production of quality olive oil and its evolution into multifunctionality is what enabled the preservation of this landscape and this multifunctionality can only be matched by a diversity of scientific studies. This study aimed at providing a foundation for the identification of the Cres landscape values within the scope of better defining further planning and management.

**Author Contributions:** Conceptualization, T.K., G.A. and M.V.; methodology, G.A. and T.K.; software used, QGIS and ESRI; writing—original draft preparation, T.K. and G.A.; writing—review and editing, G.A. and M.V.; visualization, G.A. and T.K.; supervision, G.A. and M.V. All authors have read and agreed to the published version of the manuscript.

**Funding:** This paper is part of the PhD thesis "Valorisation of the dry stone wall heritage of the Cres-Lošinj archipelago". Research funded by CARIPARO Foundation and conducted by the PhD Candidate Tanja Kremenić.

**Data Availability Statement:** Not applicable.

**Acknowledgments:** We thank the representatives of the Agricultural Cooperative Cres (Franko Fučić) and Cres agriculturists (Stefano Koljevina, Giorgio Karvin, Alberto Dunković, Frane Cesarić) for informative conversations. We thank Helena Miholić and Michela Trevisan for their help with GIS, Anita Trojanović for the contribution to the graphic elaboration of the researched area, and Alexandra Glavaschi and Tara Dmitrović with their help in proofreading.

**Conflicts of Interest:** The authors declare no conflict of interest. The funders had no role in the design of the study; in the collection, analyses, or interpretation of data; in the writing of the manuscript, or in the decision to publish the results.

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
