# Peer review of "How Did Sheep Save the Day? The Role of Dry Stone Wall Heritage and Agropastorality in Historical Landscape Preservation. A Case-Study of the Town of Cres Olive Grove"

_land, doi:10.3390/land10090978_

Round 1
Reviewer 1 Report
The article describes and tells a specific territory, in its dynamic of socio-historical-agro-cultural evolution. An excellent job of researching information, historical maps, analyzing the signs of change has been done. The processing performed is limited to the photo-interpretation of historical aerial photos, past and present land use and related dynamics.
The article tells the landscape, the writing is fluid and interesting, and full of particular uses and devices.
In addition to describing the environment and the landscape well, it does not bring news or analyzes based on relevant scientific aspects. A typical part of a scientific paper is missing, therefore the analysis of certain variables useful to demonstrate the arguments supported. From the formulation of the article it is clear that it is not the author's aim to produce evidence, but to describe a virtuous case of resilient agriculture.
The work is well done but it is far from being a scientific paper, with data and analysis to support the discussions. In this case, even if done very well, the only tangible support for the discussions is the photo-interpretation of current and historical orthophotos. Specifically concerns the place under study. It does not produce new knowledge to be applied to other case studies, or to better understand the specific dynamics that affect this area.
it's well written, the only error I found is in line 509 - correct "1)" term ... no "t)". The images are beautiful and understandable, the bibliography is appropriate. I have no particular revisions to propose. I leave it to the editor to decide whether to deem the article worthy of being published as a scientific paper.
Author Response
Document attached.

Reviewer 2 Report
I especially appreciate the successful attempt to study a very characteristic landscape of the island of Cres. The description is very accurate, the result of a good knowledge of the place and the people who live there (and who describe it). I also value the desire to achieve the conservation of this landscape. I also think that the typologies presented may be of interest for studying other Mediterranean landscapes that have similar characteristics. The only aspect I miss — though I don’t want it to be seen as a critique — is to have more data on what this landscape was like between Roman and modern times. The authors talk about resilience of agricultural and livestock structures and about the remarkable dry stone structures, but it seems as if it all started in the 16th century. Leaving this aside - I am a medievalist - I find it a good work, which deserves to be published.
Author Response
Document attached.

Reviewer 3 Report
This paper presents a very interesting dataset on the morphologies and spatial patterns of the agriculture-related dry stone structures of the area of Cres, in the Croatian Adriatic. The aim of the paper is to deliver knowledge useful for the management of this area from a perspective of preservation of the traditional agriculture practices.
In my opinion, there are some flaws that require further work for making this manuscript suitable for publication in Land.
In first place, the authors rely on the term “resilience” for summarizing the capacity of the agricultural system for surviving and adapting through time. They however do not give details on the factors that have made the system resilient and how (i.e. what processes were involved). In fact, the term itself is not sufficiently explained with respect to this specific area: how the authors recognize resilience and adaptation in this particular case, which components of the landscape are considered essential for the identity of the system so changing them would constitute a change of regime, and when in history those essential components appeared (so the statement that the landscape has been resilient through history is backed up). The authors say that the objectives of the paper are studying: “two main pillars of landscape resilience: 1) dry stone wall structures and 2) the agro-pastoral practices” (lines 92-93), but it is not demonstrated that these landscape components are actually providing resilience. Explaining what characteristics of the dry stone walls system and of the specific agro-pastoral practices are resilience providers by relating them with environmental, cultural and socio-economic factors would be needed for a meaningful analysis of the evolution of the system and its historical resilience.
I think, however, that the mention to resilience in this manuscript is a bit artificial, so not needed, and that the dataset presented in this work has a enough importance and potential of providing an interesting outcome, so fashionable terms are not needed for making it valuable.
In this regard, I think that the paper would be very much improved by paying a bit more of attention to the discussion of the dry-stone structures in terms of their function and in relation to environmental and social factors, in a sort of ecosystem services analysis, that would inform on the cultural-ecological interlinks of the physical and ethnographic characteristics of the system. Important aspects to have into account would be: topography, geomorphology and pedology in relation to the capacity of the dry-stone walls for helping to avoid soil erosion —so avoiding land degradation— and for enhancing soil fertility; land tenure and governance (including water), that would relate to socio-cultural aspects of land use and may have been a factor conditioning the size and shape of the structures and their spatial pattern; and ecological diversity in relation to topographical and pedogenetic aspects and agricultural management. It would be convenient to have into account also chronological and paleoclimate factors, since they may have been important for the three mentioned aspects.
I am obviously not asking the authors to do a totally different paper addressing my suggestions, but I do think that the manuscript would be largely improved if these aspects were considered in the discussion, which in its current form is, in my view, dismissing the potential of the data provided in the Results part, and insufficient for supporting the claims made in the Conclusions section. Information on topography, geomorphology, soils and paleoclimate is easily available. On the other hand, historical documents that contain information useful for addressing socio-cultural factors are available too, as it is shown in the introduction of the manuscript.
Regarding the chronology, in the absence of archaeological dates that unequivocally inform on the system’s age, I suggest the authors to put forward a discussed hypothesis on the origins and evolution of the system and to provide context for it in light of the age of other dry-stone terraced areas of Adriatic Croatia and Mediterranean Europe (the information that is currently presented in the introduction part —lines 59-80— can be used to this purpose, in addition to studies of other areas).
Please find some detailed comments below:
Introduction:
Lines 50-51. I would not say that Mediterranean weather has “radical” changes, and for sure soil is not scarce.
Lines 63-80. Remove the mentions to other manuscript sections
Lines 89-90. The authors’ opinion on the national and international importance of the system needs to be better justified
Line 92: “This study focuses”
Lines 95-96. This sentence implies that the authors consider that the agricultural system is and is resilient and sustainable, which has not been proved, so it cannot be taken as a premise.
Material and Methods:
This section gives details on the sources of information considered, but not on how that information has been dealt with. Please explain further.
Line 103. It is required a reference for the statement that “the study encompasses the full extent of the historically cultivated area”.
Line 125. Which is the specific time period covered by that documents?
Lines 128-129: “Establishment of a GIS database involved collection, 128 updating of existing and creating new spatial data.” is not a sentence. Rephrase.
Line 131.The “temporal land use analysis” provided in the results section is only limited (from 18th century onwards).
Line 143: remove mention to Acknowledgements section
Results:
Lines 154-156. Reference needed.
Lines 158-160. Not clear. Rephrase.
Lines 154-160. It is beyond my capacities to evaluate the accuracy of this statement based on the references provided [43
], since it is in Croatian (??), but I would like to highlight that the fact that dry stone terracing is not acknowledged in written sources does not mean that dry stone agriculture did not exist before 15th century. There are a number of examples throughout Europe in which terraces are in fact older than acknowledged in historical documents, because written sources tend to gather only the elite’s and not the peasants’ histories.
Lines 174-235. It would be great to have some info of the evolution of the Cres area compared with Croatian mainland and with other parts of Europe.
Lines 185-187. Consider mentioning climate factors
Lines 208-210. The sentence: “in a more distant area in the northern part 208 of the island (Bok) in 1920s, members of the Agricultural Cooperative decided to apply 209 this “olive-sheep” model around the town of Cres in 1956 [46].” should be a separate one.
Lines 228-229. Rephrase.
Line 236. Consider merging this subsection with 3.1 and 3.2, since they deal with past land use and management, and with 3.4 for the present day part
Line 237. Citation for “Fortis”.
Lines 246-264. I understand that the data presented in this paragraph are not results from this work, so references are needed.
Lines 318-324. Mention UNESCO’s declaration of “Art of the dry stone walling” as intangible heritage in 2018.
Line 339-341. Rephrase.
Line 345. Replace “Such walls with the” by “When their”
Line 416-426. This part should be moved to the discussion section
Line 438. Replace “vertical” by “following the slope”.
Lines 438-449. Move this info to the discussion part, and add info on the functionality of each structure and/or pattern in relation to environmental and /or socio-cultural aspects. Compare with morphologies and patterns reported for other European areas.
Discussion.
In my view, it is the most problematic section of the paper. Some resilience-related important aspect are not sufficiently discussed and the (very interesting) data provided in the results section, which in my view should occupy the largest part of the discussion, are neglected.
Lines 453-455. In this sentence, “resilience” is identified with “preservation”, which is not correct. In fact, resilience involves change, so it is the opposite to preservation. Here it would be useful to discuss which features the authors consider “essential” for defining this landscape, and which part of the “capital” should be preserved for not considering a change in regime (following Holling’s resilience theory). Also, for having resilience as a central concept in the discussion, an assessment of the remember and revolt processes should be required. This links with the statement of lines 517-520 (Conclusions section) on the influence of factors external to the Cres system. If external factors have been detected, please enumerate them, provide details on how they have been identified, and discuss what are their effects in terms of resilience provision.
Line 480. The system is repeatedly characterized as “rural-urban” in this manuscript, but no details on what the authors mean with this are given, nor on the degree of mixing between rural and urban characteristics with spatial specificity (e.g. providing maps or other info on how the rural and urban uses are distributed spatially from a totally urban to a totally rural setting and the connections between the two)
Lines 484-486. This is very interesting, I think this subject deserves more attention but, for this, the results have to be discussed more in depth in terms of the function of the structures, of the advantages or disadvantages of the variability of structures in relation to environmental characteristics, and of landscape multifunctionality and the sustainability of the practices in terms of land degradation. An estimation of the extent of the degraded area would also be highly beneficial for this discussion.
Lines 488-489. Rephrase.
Line 490-491. Which is the “contemporary meaning and value of a living agricultural 490 landscape”? Please discuss.
Line 500. Rephrase.
Lines 501-503. I agree, but this has to be discussed using the data provided in the results section: differences in patterns in relation to environmental factors, variability in structures morphology and their relation to ecological and socio-economic values, considering traditional social organization and share of resources, land tenure and governance, etc., following a sort of an ecosystem services approach.
Conclusions:
Line 513. The environmental aptitude of the Cres system is not addressed in this manuscript
Line 517-520. This statement introduces the political ecology perspective, which has not been mentioned before (it should have been addressed in the discussion, see comment of lines 453-455).
Lines 524-525. This would be, in my view, the most interesting topic to discuss in this paper
Lines 533-535. No evidence has been provided for this
Lines 538-540. Factors providing resilience and the presence or degree of resilience in this landscape have not been addressed. The possibility of reaching tipping points or passing environmental thresholds in the future has not been evaluated.
Lines 542-545. Ecological aspects and the foundations of resilience are not addressed in this paper.
Author Response
Document attached.

Reviewer 4 Report
- Line 65, is the reference to paragraph 3.1. necessary?
- Line 67, idem to paragraph 3.2.
- Line 68 and 69, idem to paragraph 3.5., 3.2 and 3.4.
- Line 74 idem to paragraph 3.4.2.
- Line 80: where is “agricultural system of its main confounds”. Wrong word? Maybe "foundations"? I don't understand
- Line 159: where is “(See: [43]), most intense transformation of the Cres”. Should be something like “(See: [43]), i.e., the most intense”
- Please see overall figure caption: source is not referred the same way in all of them. Please make it the same way.
- Line 213: where’s “(y. 1910 : inh. 4196, y. 1945 : inh. 3087, y. 1953 : inh. 1670 [42] (p. 309).” Should be: “(y. 1910 : inh. 4196, y. 1945 : inh. 3087, y. 1953 : inh. 1670) [42] (p. 309).” (lack of “)”)
- Line 277: the mentioned Figure 13 and 14 are in bold
- Line 326: the title of 3.5, “Typology of dry stone structures, buildings and patterns” doesn’t reflect the subsequent matter as “buildings” are included in “typology of dry stone structures”. So, the title must be “Typology of dry stone structures and patterns”
- Line 339: where is “Locally, the term is barbakan, has more meanings “, please write something like “Locally, the term is barbakan, which has more meanings “
- Please review all the figures’ numbers:
- in figure caption. Example: line 377 and 381: you have the Figure 21 and 22 in line 377 and the next figure is figure 24, line 380. Where is the figure 23?
- In the text:
- not all figures are referred in the text! Example: Figure 15 (line 368) is not referred in the text, as well as figure 25 and 26, line 385.
- Figures are wrong referred in the text. Example: Line 341: where is “Figure 16” isn’t Figure 15? The same to subsequent references
- Line 359: end the sentence with end point and not semicolon (;).
- In the footnote, after line 372, second line, you have forgotten a ): where is “(See: [15] (p. 145).”, must be “(See: [15]) (p. 145).”
- Line 394: you referred the figure 28, but is it right? I think the number or the figure are not right
- Line 399: figure 27’s caption without the source
- Line 411: here left something to introduce the types of crop enclosures
- Lines 413, 414 and 415: there are 4 types in figure 28, so, please, include the forth type here
- Lines 416-425: 1d type is not referred in the text
- Lines 507-510. Some words have 2 hyphens, like insufficiently in lines 507-508; management, in lines 508-509; traditional, in lines 509-510.
- Line 509: where is “considering t) the urgency” must be “considering 1) the urgency”
- Some Figures and figure’s caption are apart. Example: Figure 9 and 10, line 225, and figure caption, line 226, are in different pages;
- Please confirm reference [5], line 584. It has not enough data, for example, editors, town, etc.
A suggestion: I think it would be better if you could propose the classification of this region´s heritage as a GIAHS (like Barroso Agro-sylvo-pastoral System or The Agricultural System Ancient Olives Trees Territorio Sénia) and, even better, if you could propose it as World Heritage by UNESCO (like Landscape of the Pico Island Vineyard Culture or Cultural Landscape of the Serra de Tramuntana), as well.
This would improve the recognition from the part of the population itself as they see their heritage recognized. Very good reading and interesting subject.
Author Response
Document attached.

Round 2
Reviewer 3 Report
The current version of the manuscript shows a good improvement compared to the earlier version. The discussion is now of higher interest, as it is more focused in landscape multifunctionality and management, and puts more emphasis on the analysis of the different grove patterns developed as a result of the interplay between anthropogenic and environmental factors.
Nonetheless, the discussion part, which is the one on which I have focused the majority of my detailed comments in the document enclosed, requires, in my view, a bit more of effort in stressing the key messages, which remain somehow hidden and not fully exploited. Also, the discussion part needs to support a fair part of its claims and arguments with references, or with further discussion if references are not available. I have highlighted in the enclosed document some of them.
I suggest also to include a mention to multifunctionality and management in the conclusions part, since these are, in my view, important outcomes of this work and should be more emphasized.

Author Response
pdf attached
